# Utility of Exosomes in Ischemic and Hemorrhagic Stroke Diagnosis and Treatment

**DOI:** 10.3390/ijms23158367

**Published:** 2022-07-28

**Authors:** Eun Chae Lee, Tae Won Ha, Dong-Hun Lee, Dong-Yong Hong, Sang-Won Park, Ji Young Lee, Man Ryul Lee, Jae Sang Oh

**Affiliations:** 1Department of Neurosurgery, College of Medicine, Cheonan Hospital, Soonchunhyang University, Cheonan 31151, Korea; lec9589@gmail.com (E.C.L.); madeby58@gmail.com (D.-H.L.); dydehdghd@gmail.com (D.-Y.H.); ppphilio3@gmail.com (S.-W.P.); applesori82@gmail.com (J.Y.L.); 2Soonchunhyang Institute of Medi-Bio Science (SIMS), Soon Chun Hyang University, Cheonan 31151, Korea; htw5200@gmail.com

**Keywords:** extracellular vesicles, exosome, stroke, ischemic stroke, hemorrhagic stroke, exosome isolation

## Abstract

Stroke is the leading cause of death and neurological disorders worldwide. However, diagnostic techniques and treatments for stroke patients are still limited for certain types of stroke. Intensive research has been conducted so far to find suitable diagnostic techniques and treatments, but so far there has been no success. In recent years, various studies have drawn much attention to the clinical value of utilizing the mechanism of exosomes, low toxicity, biodegradability, and the ability to cross the blood–brain barrier. Recent studies have been reported on the use of biomarkers and protective and recovery effects of exosomes derived from stem cells or various cells in the diagnostic stage after stroke. This review focuses on publications describing changes in diagnostic biomarkers of exosomes following various strokes and processes for various potential applications as therapeutics.

## 1. Introduction

Stroke is a devastating neurological disease associated with high mortality and disability [1]. Strokes can be divided into two groups: ischemic strokes, caused by the blockage of blood vessels, and hemorrhagic strokes, caused by the rupture of blood vessels [2]. The main treatment goal following an ischemic stroke is to restore blood flow as soon as possible after symptoms develop. For hemorrhagic stroke, treatments involve the removal of intracranial blood clots or intraventricular blood through invasive surgery and the management of intracranial pressure to effectively reduce mortality [1,3]. The two main approaches to achieve patency are intravenous thrombolysis and intravascular intracranial thrombectomy; both of these treatments have a narrow effective window [4,5]. Recently, exosomes have attracted interest due to their potential stroke-diagnostic and therapeutic applications. Exosomes are a subset of extracellular vesicles (EVs) ranging in diameter from 30 to 150 nm, and exosomes secreted from various exogenous cells and organs have well-documented potential neural regenerative roles in stroke [6,7,8,9]. Compared to cell-based therapies, some stem-cell-derived exosome-based therapies reduce potential tumor and immunogenic side effects while exhibiting similar protective effects. The underlying mechanism of the exosome therapeutic effect is primarily the delivery of molecules, particularly microRNA (miRNA) [10]. Numerous studies have demonstrated that EVs can mediate cell–cell/tissue communication, primarily by delivering cargo such as proteins and miRNAs [11,12]. The development of multifactors from brain injury in post-stroke patients is not only a potential source for identifying diagnostic markers but also a useful tool for discovering new treatment protocols. Based on previous studies, this review highlights the role of exosomes in stroke injury. An overview is presented of the current knowledge on the subject and the diagnostic and prognostic roles of exosomes in clinical trials in post-stroke patients [13]. A better understanding of the role of exosomes in post-stroke neurodegeneration may contribute to faster stroke diagnosis, prediction of stroke outcome and prognosis, development of new therapies, improved patient care, and reduced medical costs.

## 2. Characteristics of Exosomes

Exosomes are small EVs between 30–150 nm in size secreted by all cell types and are responsible for intercellular communication [8]. Exosomes are the smallest category of EVs and are released from multivesicular bodies (MVBs). As endogenous nanovesicles, they play a key role in cell-to-cell interactions by transferring functional genetic information and proteins [14,15,16]. Exosome biogenesis starts with the development of early endosomes, which develop into late endosomes and then MVBs [17] (Figure 1). Endocytic vesicles are delivered to nascent endosomes located at the outer edge of the cell, which alters the protein material of the vesicle and thus converts it into late endosomes. MVBs are derived from late endosomes budding back into the endosomal lumen, whereas later endosomes release exosomes. MVBs are mediated by actin, microtubule scaffolds, and specific components of the Rab family, which allow the continuous transportation of MVBs and their eventual attachment to the plasma membrane and secretion of exosomes. The secreted exosomes are moved to the recipient cell followed by receptor–ligand interaction, fusion, and endocytosis. Exosomes are composed of a lipid bilayer membrane structure and carry functional contents including RNA (mRNA, miRNA, and other RNAs), DNA, proteins, lipids, and metabolites [18,19,20] (Figure 1). Exosomes were originally identified in sheep reticulocytes [21,22]. Since their identification, numerous studies have showed that exosomes can be extracted from almost all body fluids including blood, cerebrospinal fluid (CSF), urine, saliva, amniotic fluid, and breast milk. As exosomes can be extracted from various cell types, exosome isolation methods can be varied. The most common markers of exosomes are tetraspanins (CD9, CD63, and CD81), MVB-related endosomal sorting complexes required for transport (ESCRT) proteins (Alix and TSG101), and heat shock proteins (HSPs; HSP60, HSP70, HSPA5, CCT2, and HSP90). Exosomes also contain numerous functional RNA molecules (e.g., miRNAs) that reflect the physiological and pathological properties of the cell of origin. Exosomes release mature miRNAs into recipient cells that regulate gene expression and various cellular and molecular pathways [23]. A recent study demonstrated that exosomal miRNAs are involved in the regulation of physiological and pathological processes after ischemic stroke and contribute to brain remodeling by transporting cargo [24]. Therefore, exosomes have been considered promising biomarkers for the early diagnosis and prognosis of stroke as well as prospective drugs for stroke treatment [25]. 

## 3. Exosomes in Stroke

### 3.1. Role of Exosomes in Ischemic Stroke

Ischemic stroke is a disease caused by the insufficient supply of blood and oxygen to the brain and is mainly caused by the stenosis of the intracranial artery and the occlusion of the middle cerebral artery [26]. When an embolism or clot blocks a cerebral artery, circulating platelets are rapidly recruited to the area of the blocked vessel. Platelets along with thrombin and fibrin are major contributors to thrombosis due to tissue-factor-activated blood coagulation in occluded arterial sites. Cerebral ischemia induces a variety of secondary effects including reperfusion injury, ischemic hypoxia, increased intracellular calcium levels, blood–brain barrier (BBB) damage, reactive oxidative stress, apoptosis and inflammation, and ionic imbalance, ultimately leading to neuronal damage [27]. During ischemic stroke, brain tissue ischemia and hypoxia interfere with mitochondrial oxidative phosphorylation, leading to impaired energy metabolism and decreased ATP production. Tissue ATP is rapidly depleted and calcium ions (Ca^2+^) are released from mitochondria and the endoplasmic reticulum [28]. This process produces free radicals that constrict large blood vessels, exacerbating ischemia and hypoxia and delaying recovery. In addition, during ischemic stroke, ROS incidence rises from a normal low level to a peak during reperfusion, contributing to a potential acceleration of apoptosis or cell necrosis [29]. Notably, cells in the peri-infarct region are still alive in the early stages of ischemia. However, most of these will undergo cell death due to the unsuitable microenvironment. Furthermore, the destruction of the blood–brain barrier (BBB) during ischemic stroke can lead to brain edema formation and hemorrhagic deformation [30]. At this stage, the repair and remodeling processes are initiated, including angiogenesis, neurogenesis, synaptogenesis, recovery of the BBB, and oligodendrogenesis [31]. Newly formed neurovascular units can partially replace damaged tissue and improve outcomes [32]. Exosomes are produced by brain cells following brain injury and trigger various responses. Some exosomes appear to have neuroprotective and nerve-repairing effects. In addition, these exosomes can cross the BBB and circulate in peripheral blood and cerebrospinal fluid and can be excellent non-invasive biomarkers for ischemic stroke diagnosis and prognosis [33]. However, some exosomes are associated with neurodegeneration [34,35]. Recent studies have uncovered numerous components in circulating exosomes that may serve as biomarkers for ischemic stroke, specifically miRNAs.

### 3.2. Role of Exosomes in Hemorrhagic Stroke

A hemorrhagic stroke occurs when a blood vessel ruptures and bleeds into the brain. Within minutes, brain cells begin to die. Causes may include intracranial aneurysms, arteriovenous malformations (AVMs), and ruptured arteries or veins [36]. Hemorrhagic strokes have a relatively lower incidence than ischemic strokes.

Among the symptoms of hemorrhagic stroke, seizure symptoms associated with the head trauma that may occur during hemorrhagic stroke onset can potentially exacerbate the bleeding, and with localized brain damage, intracerebral hemorrhage occurs within the cerebrum [36,37]. If a craniectomy is performed following a hemorrhagic stroke, seizures may damage unprotected parts of the brain, resulting in further damage. However, it is sometimes difficult to identify the bleeding location as the blood often moves in a cluttered manner. Since subcortical hemorrhages migrate to encapsulate the cortex, it can be difficult to identify the origin. Therefore, patients with cortical or subcortical bleeding are more likely to develop stroke-related seizures than patients with other types of bleeding. The BBB is damaged as soon as a hematoma appears. After local bleeding and brain injury, edema, leukocyte influx, and potential neuroactive agents penetrate the CSF and blood through the broken BBB [38]. After the initial breakdown of the BBB, toxic free radicals, proteins, lipids, miRNAs, and exosomes are released through the bloodstream. Thus, the utilization of exosomes as circulating biomarkers might provide a dynamic and powerful tool for disease screening, diagnosis, prognosis, and monitoring of therapeutic efficacy [39]. Unfortunately, none of these have reached clinical practice yet.

## 4. Exosome Extraction Methods

### Extracting Exosomes from the Blood Serum and CSF of Stroke Patients

Exosomes contain both proteins and nucleic acids from the cell of origin and can be released through the blood serum. When isolating exosomes from blood serum, they must first be separated without loss into purified biologically active vesicles [40,41]. There have been many recent studies on methods of extracting exosomes from blood serum. When selecting the most suitable technique among various exosome extraction methods, it is necessary to consider the concentration, purity, and size of exosomes and exosome RNA as a technique that can extract with the highest extraction efficiency and purity. Exosome isolation techniques can be separated using Ultracentrifuge (UC), ExoQuick, or Total Exosome Isolation Reagent (TEI) [42]. UC had the lowest particle recovery but the highest protein purity in the exosome isolation step. However, Exoquick and TEI, which are commercial kits, showed higher extraction efficiency than the UC method. Size exclusion shromatography (SEC) techniques can also be used, but the yield of vesicles is low using SEC. In blood serum, the removal of albumin, a soluble plasma protein, is important, and there are inherent limitations in isolating exosomes from serum, so it should be evaluated carefully and by applying appropriate experimental techniques [43].

Cerebrospinal fluid-derived exosomes are considered biomarkers that can be utilized for the early molecular diagnosis of neurological diseases. The most important step in the separation of exosomes from CSF is the removal of cell debris [44]. However, the method of isolating high-purity exosomes from CSF is still in its infancy and has many limitations in its clinical application. The technique called EXODUS has a shorter separation time and higher particle yield than basic separation techniques such as UC [45]. More research needs to be conducted on techniques with efficient and fast approaches such as these techniques.

## 5. Relationship between Exosomes and Mechanisms Associated with Stroke

### 5.1. Association between Exosomes and Angiogenesis

Angiogenesis involves the modulation of basal lamellar matrix separation and remodeling, as well as endothelial cell proliferation, migration, and elongation, with the participation of pericytes to form new micro-vessels. Stroke induces the overexpression of matrix metalloproteases, which are important for basal matrix segregation. In addition, ischemia promotes the upregulation of integrins and angiogenic factors, including vascular endothelial growth factor (VEGF), platelet-derived growth factor, fibroblast growth factor (FGF), angiopoietin-1/-2, and Tie-2 [46]. It can activate cells and promote angiogenesis. Moreover, endothelial progenitor cells are motivated to participate in angiogenesis. Exosomes can transport therapeutic agents through membranes, thus allowing for the involvement in both autocrine and paracrine signaling via the delivery of proteins and RNAs essential for angiogenesis that occurs under ischemic conditions [47,48]. Proteins and miRNAs are key regulators, highly expressed in exosomes, that promote angiogenesis and intracellular communications, which ultimately affect growth factors and gene expression [49,50] (Figure 2). 

The effect of angiogenesis has been confirmed by estimating the expression of angiogenic growth factors and the upregulation of miRNAs in an ischemic myocardial model [51,52]. Exosomes can promote angiogenesis by upregulating angiogenesis-promoting molecules and downregulating anti-angiogenic molecules [53]. Another study has suggested that exosomes can be secreted from cardiac progenitor cells, endometrial stromal cells, and human pluripotent stem cells to promote angiogenesis [54,55]. Additionally, in the mouse hindlimb ischemia model, exosomes derived from iPSC-derived mesenchymal stem cells (MSCs) induced angiogenesis by increasing the microvascular density [56]. Exosomes secreted from human adipose-derived MSCs promote angiogenesis, resulting in an increase in vessel length and the number of vascular branches (Figure 2). These studies support the use of exosomes as potential angiogenesis-promoting therapeutics following various strokes.

In hemorrhagic strokes, small arteriole ruptures occur due to vascular pathological changes. Following vascular rupture, blood vessels remain vulnerable, which represents a disadvantage in vascular pathologies, including vascular remodeling and plaque formation [57]. After blood vessel rupture, hemoglobin enters the brain and induces pathological changes such as edema, inflammation, and apoptosis, eventually leading to neurological deficits. One study confirmed the ability of exosomes extracted from endothelial progenitor cells to promote angiogenic repair and functional restoration [58].

An endothelial cell-specific miRNA, miR-126, was also recently confirmed to induce angiogenesis following intracerebral hemorrhage (ICH) [33,59]. We confirmed that miR-126 promotes angiogenesis following ICH in response to angiogenic growth factors including VEGF or fibroblast growth factor [60,61]. The overexpression of miR-126 further supports its potential angiogenic effect (Figure 2).

### 5.2. Association between Exosomes and Neurogenesis

Along with angiogenesis, neurogenesis is an important process in stroke recovery [62]. Various studies have shown that exosome-based therapy contributes to the transformation of neural stem cells and promotes neurogenesis [54,63]. MiR-124 is abundantly expressed in brain tissue and plays an important role in neurogenesis. Its overexpression can lead to neuronal differentiation, and it was shown to be upregulated in the ischemic region after middle cerebral artery occlusion (MCAO) [64].

Another study showed that stroke increases neurogenesis in MCAO models and increases the number of neural progenitor cells as well as neuroblasts, which have also been observed in the brain of ischemic patients.

Stroke-induced neural stem cell proliferation in the ventricular–subventricular zone (V/SVZ) cleft is associated with activated cerebral endothelial cells [65,66,67]. Newly generated neuroblasts in the V/SVZ migrate along the cerebral blood vessels to the infarct surrounding area [68]. Additionally, neurogenesis following stroke is linked to angiogenesis [69,70,71,72]. Thus, the blockade of cerebral angiogenesis damages neurogenesis in the ischemic brain.

Neuroblasts are expected to play a functional role in the brain recovery process, as the depletion of neuroblasts exacerbates the recovery process as well as the neurological outcome during stroke recovery [73]. Studies have shown that following such a phenomenon, exosomes from CSF and neural stem cells promote neural stem cell function and immune system function by regulating the intercellular pathways. CSF exosomes containing both proteins and miRNAs activate the IGF/mTORC1 pathway in neural stem cells and promote neural stem cell proliferation (Figure 2) [74].

Intranasal administration of human bone marrow mesenchymal stem cell-derived exosomes (A1-exosome) with anti-inflammatory properties for status epilepticus following hemorrhagic stroke has been shown to reduce the loss of glutamatergic and GABAergic neurons and limit the hippocampal inflammation [75,76,77]. In addition, the neuroprotective and anti-inflammatory effects were weakened in the vehicle-administered group; moreover, a persistent inflammatory response and functional deficits were present. In contrast, normal hippocampal neurogenesis and long-term maintenance of cognitive and memory functions were achieved in the nasally administered A1-exosome group (Figure 2).

### 5.3. Association between Exosomes and Autophagy

The autophagic process is underlined by a catabolic pathway in which long-lived proteins, damaged organelles, and misfolded proteins are degraded and recycled for the maintenance of cellular homeostasis and normal cellular function [78,79]. Autophagy plays an important homeostatic role in cell survival regulation. Accumulating evidence shows that autophagy is activated in various cell types in the brain, such as neurons and glial cells, during ischemic stroke. A basal level of autophagy is essential for proper neuronal function and allows neurons to persist. A moderate increase in autophagic activity can occur in mild conditions of ischemia, hypoxia, or nutrient deprivation. In more severe stages, such as acute ischemic stroke, it can lead to an increase in autophagic flux, resulting in autophagic cell death, thus, indicating the strong relationship between autophagy and apoptosis [80].

In a recent study, astrocyte-derived exosomes (AS-Exo) were isolated, and their effects on MCAO were confirmed by intravenous injection into mice via the tail vein. AS-Exo were shown to improve nerve damage by controlling autophagy, which was shown using TCC and TUNEL staining of mouse brain sections. The immunohistology results confirmed that AS-Exo suppressed the expression of GFAP and Iba-1 in the MCAO model, and the expression of Beclin-1, LC3-I, LC3-II, and P62 was also suppressed in the AS-Exo-administered group [81,82,83].

A different study suggested that autophagy reduction following stroke in a mouse model can be achieved via exosomal delivery of miR-25-3p from adipose-derived mesenchymal stem cells (ADMSC) to target cells [84] (Figure 2). Inhibiting autophagy completely cannot have exclusive therapeutic effects [85]. A study focusing on the HIF-1a pathway that induces either P53 or BNIP3 expression in stroke provided evidence that DMSC-EV exhibits neuroprotective effects and enhances neuronal recovery by inhibiting ischemia-induced autophagy. The inhibition of autophagy is in turn mediated by EVs delivering ADMSC-derived miR-25-3p to target cells [85], the latter interfering with the p53/BNIP3 signaling pathway. Studies have shown that these new observations of stem-cell-derived EVs may be used for the development of novel therapeutic targets and strategies for the treatment of ischemic stroke.

Inhibition or knockdown of PDE1-B significantly enhances autophagy flux, promoted M2 and suppresses the M1 phenotype in BV2 cells. EVs derived from regulated microglia are expected to modulate the survival of cortical neurons in ischemic conditions. Meanwhile, EV-miRNAs that improve ischemia-induced nerve damage through autophagy regulation have been found [86].

It has been observed that these effects are dependent on circSHOC2 of IPAS-EXO, inhibit neuronal cell death, and ameliorate neuronal injury by agonizing miR-7670-3p/SIRT1 in an in vivo model by regulating autophagy. Thus, we demonstrated that exosomes secreted from ischemic prestress astrocytes (IPAS-EXOs) co-cultured with neurons exert neuroprotection [86,87].

Oxidative stress, inflammation, and free iron accumulation after ICH onset can trigger ferroptosis and autophagy. Ferroptosis is a newly identified form of apoptosis that differs from apoptosis, necrosis, and autophagy. It can exacerbate the inflammatory response by promoting microglia activation via autophagy involving the beclin-1-Atg5 and nuclear factor-κB pathways. Following ICH, autophagy may promote brain injury in the early stages, but at the same time, it may be neuroprotective at later stages due to the removal of cellular debris [88]. In a recent study, exosomes from miR-19b-3p-modified adipose-derived stem cells (ADSCs) effectively attenuate ferroptosis following ICH [89].

### 5.4. Association between Exosomes and BBB

The BBB consists of endothelial cells (EC), a basement membrane, astrocyte foot processes, and pericytes [90]. This barrier plays a fundamental role in the maintenance of normal neural function, protecting the brain microenvironment from blood flow disturbances by regulating permeability based on the stringent and stable CSF environment. Strokes are associated with BBB disruption, which begins with ischemia but continues to worsen with persistent hypoperfusion. This deterioration is mainly due to a lack of nutrients but may also be due to altered structural changes. This is evidenced by the fact that even after blood flow is restored, the BBB permeability is not restored and functions below the baseline level [91]. During a hemorrhagic stroke, the BBB is damaged as soon as a hematoma appears. In other words, the cascade following BBB disruption is a gradual degradation process, and consequently, the release of different biomarkers may change over time. According to a recent study, macrophage-derived exosomes ameliorated the impaired BBB and traumatic brain injury [92].

Exosomes are the best choice because of their natural properties, such as negligible toxicity, BBB permeability, cyclic stability, production and storage advantages, ability to encapsulate endogenous bioactive molecules, strong protection against cargo spillage, and excellent transport efficacy to distant somatic cells [93].

The ability of exosomes to cross the BBB increased the interest in and research on utilizing exosomes as drug delivery systems [94,95].

Currently, the mechanism of exosome BBB crossing remains unclear [96]. However, studies examining exosome trafficking through the BBB point to the uptake by BMECs via endocytosis, followed by fusion to the BMEC endosomes and release into the brain [90]. Interactions between exosomes and BMECs in vitro demonstrate that exosomes retain their ability to cross the BBB under healthy and stroke-like conditions [90]. We confirmed that exosomes cross the BBB and are taken up by endothelial cells in a 3D BBB static model [97].

A recent study found that intravenous treatment of human neural stem-cell-derived exosomes (NSC-EV) improved both sensorimotor function and protected the integrity of the BBB as well as with matter (WM) in a preclinical thromboembolic porcine model of ischemic stroke [98].

In a different study, treatment with multipotent mesenchymal stromal-cell-derived exosomes after the establishment of ICH in a rat model increased the density of endothelial barrier antigen-immunopositive cells, compared to controls [99]. The increased density of these cells indicates mature blood vessels in which the BBB functions [100]. Angiogenesis accelerates neurite outgrowth and synapse formation in the brain, contributing to functional recovery [101,102,103]. It was confirmed that the systemic administration of MSC-derived exosomes improved spatial learning (Morris water maze) and motor function [104].

This finding of exosomal BBB crossover has been confirmed by numerous studies. However, in an exosomal biodistribution study, after the intravenous administration of natural exosomes into the body, they were rapidly removed from the target site and accumulated in organs of the reticuloendothelial system (RES) such as the lung, liver, and spleen [105,106,107]. Therefore, there is a need to engineer exosomes with specific targeting properties before their use as therapeutic agents for stroke treatment.

## 6. Diagnostic Value of Exosomes in Stroke

The diagnosis and treatment of stroke should be completed as soon as possible, preferably before the patient arrives at a stroke center [25]. Due to the nature of the disease, the treatment effectiveness varies over time; thus, it is important to minimize time delays during the diagnostic process [108]. Stroke diagnosis relies on the ability to differentiate between stroke mimics (migraines, seizures) and actual strokes as well as between hemorrhagic and ischemic strokes. Currently, the latter is only possible following neuroimaging (computed tomography (CT) or magnetic resonance imaging (MRI). To accelerate the treatment initiation time, a mobile stroke unit (MSU) with an onboard CT scanner has been developed that can diagnose ischemic and hemorrhagic stroke in hospitals and allow treatment initiation in ambulances [109]. This early initiation of treatment is associated with improved functional outcomes compared to in-hospital thrombolysis. However, these diagnostic techniques are time-consuming and have limitations that cannot be immediately overcome. The utilization of exosome profiles for stroke diagnosis has been reported by several studies and is shown to be easily and consistently measured thus acting as an acute diagnostic tool [25]. Such tools might also allow the differentiation between ischemic and hemorrhagic strokes (Figure 3).

The plasma miRNA profiling of ischemic stroke patients and animal models has been investigated, and several differentially regulated miRNAs that may have potential diagnostic value in determining stroke severity and outcome have been identified [110]. It would be interesting to evaluate the presence of these differentially regulated miRNAs in exosomes, particularly those of neuronal origin. In addition to ribonucleic acid, several proteins such as MMP-9, S100β, ICAM1, and GFAP have been shown to have potential diagnostic value in stroke [111]. The analysis of exosomal miRNA, mRNA, and protein cargo may reveal distinct profiles compared to those detected in plasma samples. In addition, Annexin-V positive exosome showed that neural progenitor cells (CD34+ and CD56+), platelets (CD61+), ECs (CD146+), erythrocytes (CD235ab+), and leukocytes (CD45+) were isolated from acute ischemic stroke patients and shown to be increased in blood samples [112]. During early diagnosis of ischemic stroke, miRNAs, the most common exosomal contents, can be utilized. A previous study confirmed that the serum levels of miR-9 and miR-124 were upregulated during AIS. All these miRs were correlated with National Institutes of Health Stroke Scale (NIHSS) scores as well as infarct volume and serum concentrations of interleukin-6 [113]. In addition, a different study showed that the expression of miR-223, which correlates with stroke severity, and miR-134 within 24 h following stroke onset was significantly increased [113]. In contrast, levels of exosomal miR-152-3p were significantly lower in patients with aortic atherosclerosis and in patients with an NIHSS score of 7 or higher. Thus, these values can be differentiated and are more meaningful in the acute phase than in the chronic phase of stroke [114].

Therefore, to predict the cause and severity of stroke in the future, it would be crucial to re-evaluate the sensitivity and specificity of these biomarkers and use exosomes to discover completely new biomarkers.

Several cell types in the brain and circulation have been shown to release EVs into the blood during hemorrhagic stroke. These cells include nerve cells, neural progenitor cells, and blood as well as vascular cells (ECs, platelets, RBCs, granulocytes, and WBCs (including monocytes and lymphocytes)) [115]. It has been examined whether all these cells release exosomes during stroke onset. Exosomes isolated during ICH in hemorrhagic stroke patients showed temporal profiling, and exosomes released from ECs, leukocytes, and RBCs increased. In another study, the expression of miR-144-5p was high in hematoma-derived exosomes from patients with chronic subdural hematoma in hemorrhagic stroke [116]. MiR-144-5p inhibits the expression of SDC3, which can inhibit EC migration and tube formation, thereby inhibiting angiogenesis [116,117]. It has been shown that, if human ECs (HUVEC) are co-cultured with hematoma-derived exosomes, miR-144-5p increases, and ANG-1 and ANG-2 expression is altered to promote angiogenesis and cell permeability. ANG-1 regulates vascular maturation as well as EC adhesion and migration, while ANG-2 promotes the dissociation of pericytes from existing blood vessels as well as increases vascular permeability and is mainly secreted by epithelial cells in active vascular remodeling sites. Therefore, targeting miR-144-5p would be of interest for the diagnosis and treatment of chronic subdural hematoma (CSDH) patients [116]. In addition, a previous study used exosomal miRNAs extracted from the CSF of patients with MMD and using a micro-Murray analysis indicated that miR-574-5p, miR-3679-5p, miR-6124, and miR-6165 were upregulated, whereas miR-6760-5p was downregulated [118]. These findings confirm the effectiveness of exosomal miRNA differentially expressed in CSF and that the identified miRNAs can play a functional role in the pathogenesis of MMD patients.

As a result of these studies, medical findings of ischemic and hemorrhagic stroke were implemented in point of care (POC) tests using diagnostic techniques such as sequencing, polymerase chain reaction (PCR), and proteomics using exosomes as biomarkers [119,120].

## 7. Therapeutic Value of Exosomes in Stroke

Recombinant tissue-type plasminogen activity (alteplase), which was approved by the Food and Drug Administration (FDA, USA) in 1996, is the only drug to date that has demonstrated recovery results when administered to patients in the early stages of an ischemic stroke. Additionally, stroke treatment includes surgical operation [5]. Although endovascular thrombectomy can prolong the treatment period and reduce the risk of intracranial hemorrhage, there are limitations due to vessel distortion, arterial stenosis, and the possibility of thrombus inaccessibility [121]. Therefore, it is important to supplement the current limited stroke treatment by developing novel and effective treatments. Recently proposed treatment methods using exosomes include direct transplantation therapy, cell source delivery therapy, and the use of exosomes as drug carriers [3,84].

As for the method of treatment consisting in the direct administration of exosomes, its therapeutic roles have been investigated in MCAO mouse models receiving exosomes into the subventricular region. In addition, MSC-derived exosomes are widely studied for their utility in stroke treatment as they promote the restoration of nerve cell function, nerve protection, and nerve regeneration regulation [32].

Prion protein is involved in cellular defense against damage caused by hypoxia and ischemia. In one study, astrocyte-derived exosomes cultured in a hypoxia environment enhance neuronal defenses against oxidative and ischemic stress [122] (Figure 4).

According to one study, it was confirmed that determining the therapeutic potentials of curcumin-loaded embryonic stem cell exosomes (MESC-exocur) reduced vascular inflammation and broadly reduced the levels of inflammatory cytokines TNF-α and ROS [123] (Figure 4).

M1 microglia exacerbate brain damage by releasing pro-inflammatory and neurotoxic factors, and M2 microglia secrete anti-inflammatory and neurotrophic mediators to repair brain damage [124]. Utilizing this mechanism, one study confirmed that bone marrow stem-cell-derived exosomes protect microglia from pro-inflammatory neurotoxicity by inhibiting microglia inflammation and promoting M2 phenotypic polarization [125].

In a hemorrhagic stroke study, an IHC model was established by injecting 1 μL of saline containing 0.5 U collagenase type VII (Sigma-Aldrich) at the injection site coordinates from bregma, followed by a release from endothelial progenitor cells (ACE2EPCs) [126] (Figure 4). The administration of angiotensin-converting enzyme2 -overexpressed endothelial progenitor cells derived exosomes into the tail vein as a therapeutic strategy was shown to alleviate functional neurological deficits and reduce hematoma size and cerebral edema 2 days after ICH onset [127]. Edema is one of the major pathophysiological features of ICH, while the main cause of edema is damage to the BBB [128]. It has been shown that these exosomes increased the BBB permeability in vivo. The induction of the expression of formation markers demonstrated improved functional recovery of the higher canal. Therefore, further studies are needed to determine whether exosome treatment can affect brain plasticity in an adaptable and diverse way, even in hemorrhagic strokes [129].

Further, the side effects of exosome treatments as cell-based therapeutic agents should be minimized [99]. The delivery route of exosomes is also an important area of study. Intravenous, intra-arterial, intracerebral ventricle, and intracranial injections are methods that directly deliver the exosomes into the damaged area. However, direct injection causes trauma, and a single exosome administration is often ineffective; such administration methods are associated with various complications such as intracranial lesions, hematomas, and epilepsy [130]. Intravenous injections are a common transplantation approach, which is non-traumatic and convenient and causes low damage. However, depending on the blood volume and blood flow in the blood vessels, it can cause a lack of concentration of the administered exosomes. Recently, administration through the nasal route was studied. Intranasal delivery is a simple and non-invasive surgical method. It is uniquely capable of bypassing the BBB, and the transmission medium corresponds to non-nerves such as the trigeminal nerve and CSF [131]. Further developments based on these studies should minimize the side effects of exosome treatment as a cell-based therapeutic agent [132].

## 8. Future Prospects of Exosomes in Stroke Diagnosis and Treatment

The potential application of exosomes for stroke diagnosis and treatment include drug release stability, brain neuron cell reconstruction, immunomodulatory functions, quick diagnostic technique, targeted therapy, and target delivery by penetrating the BBB. Thus, future research should address these various applications [93].

Following stroke, exosomes can be synthesized and released into brain cells, cross the BBB, and be detected in peripheral blood or cerebrospinal fluid and are also released into the bloodstream from blood cells and ECs in response to stroke [3]. Exosomes are involved in increasing long-term neuroprotection after stroke, promoting nerve regeneration, enhancing neurological recovery, and regulating peripheral immune responses. They also enhance angiogenesis, neurogenesis, and remodeling of axonal dendrites [133]. Exosomes secreted by brain ECs can also communicate with brain cells (including neurons and glial cells) and terminal cells of other organs during stroke recovery to act as messengers during brain reconstruction [96]. However, in addition to these therapeutic effects, exosomes can also adversely affect end organs. The release of exosomes from damaged central nerve cells into the peripheral circulation after stroke affects other organs such as the spleen, which increases circulating inflammatory cytokine production and modulates peripheral immune–inflammatory responses by recruiting and activating T and B lymphocytes [134].

In numerous recent studies, the interest in exosomes as stroke diagnostic markers and therapeutics is increasing. A previous study showed that exosomes isolated from stem cells or other cells can be expected to have preventive and recovery effects during stroke [135]. Most studies on exosomes are in vitro and few are in vivo; their physiological effects in patients remain to be investigated. At this stage, the exosomal content, dose, function, and activity should be optimized for conditions including the patient′s age, sex, comorbidities, and other factors [4]. Stability studies using exosomes as therapeutics are also important. Indeed, potential carcinogenic features have been noted in the literature following the delivery of EVs or exosomes. Thus, although the utilization of exosomes for stroke treatment is effective for reversible and neuroprotective therapeutic approaches, studies on safety, dose–response, multi-dose, and carcinogenic potential are required [7].

More in-depth research is needed on the use of exosomes as early diagnostic markers. Exosomes can cross the BBB, and as they have a double membrane structure that protects them from ribonucleases in the blood, exosome miRNAs are stable and resistant against degradation. Therefore, exosomal miRNAs can be utilized as ideal biomarkers that can be probed from circulating body fluids [33].

## 9. Conclusions

Stroke is one of the leading causes of death worldwide. In stroke patients, the treatment effect is time-dependent; thus, it is crucial to minimize the treatment delay.

Exosomes are released from almost all living cells and play an important role in intercellular communication with their small particle size. By utilizing these characteristics, exosomes can be used to develop biomarkers for stroke diagnosis and therapeutic agents that pass through the BBB. An exosome-based diagnostic kit for ischemic and hemorrhagic stroke can be developed and used for acute POC diagnostic testing. Therefore, the studies analyzed here can contribute to stroke diagnosis before patients arrive at the hospital and ultimately lead to improved treatment for stroke patients. However, the development of diagnostic technologies that can be obtained at a low cost and on a large scale is still in its infancy and needs to be addressed. Due to these limitations, the research stage linking exosomes and stroke is still in its infancy. Nevertheless, it is necessary to conduct large-scale population studies analyzing the differences in exosome release and various exosome isolation methods according to cell culture conditions and donors. Optimized research methods must be derived at each research step, and future studies aimed at identifying suitable cells that generate exosomes are needed.

## Figures and Tables

**Figure 1 ijms-23-08367-f001:**
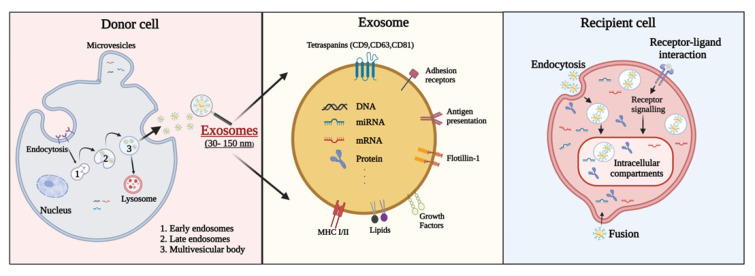
Exosome surface, components of exosome secretion, and interactions between extracellular vesicles (EVs) and recipient cells. EV membrane proteins can be delivered to receptor molecules on the surface of recipient cells, where they trigger signaling cascades. EV membrane proteins can also be cleaved by proteases and converted into soluble fragments that serve as ligands for their cell receptors. EVs can fuse with the membrane of the recipient cell to transfer cargo to the recipient cell. Migrated exosomes can be activated in the recipient cell in several ways. The secreted exosome fuses to the target cell and assists receptor–ligand interaction and endocytosis.

**Figure 2 ijms-23-08367-f002:**
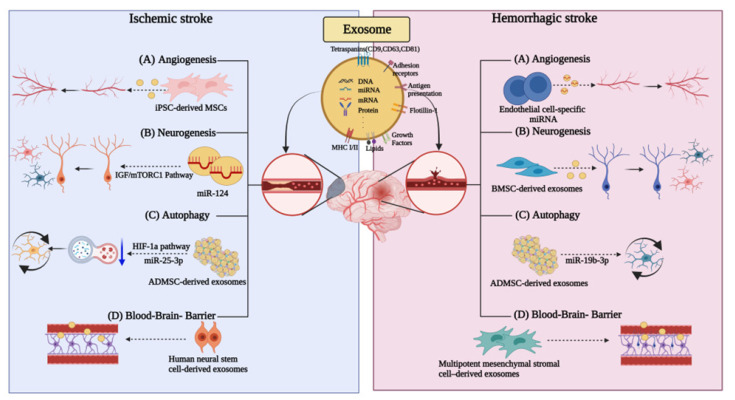
Exosomes are involved in different ways in ischemic and hemorrhagic strokes. Different roles of exosomes in (**A**) angiogenesis, (**B**) neurogenesis, (**C**) autophagy, and (**D**) blood–brain-barrier in ischemic stroke and hemorrhagic stroke.

**Figure 3 ijms-23-08367-f003:**
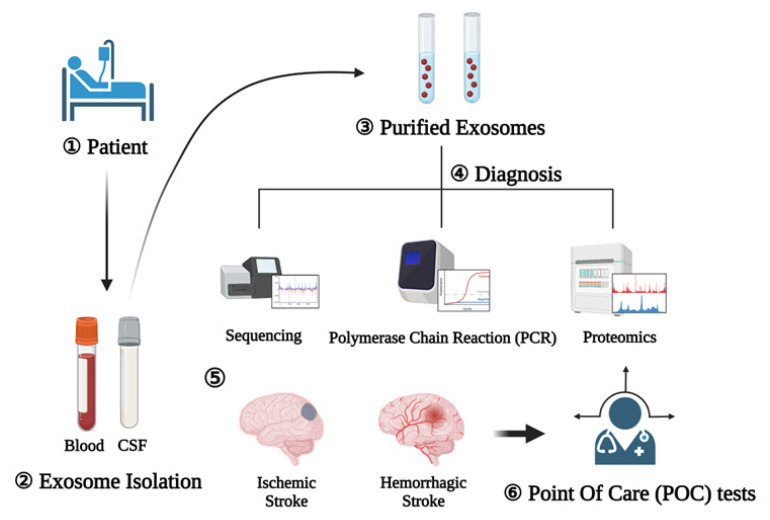
Exosomes as diagnostic markers (1). Blood and CSF are collected after the patient develops (2). After that, exosomes are extracted (3) and diagnosis is performed through sequencing, polymerase chain reaction, and proteomics (4). Schematic representation of the procedure for using exosomes as diagnostic markers for the rapid diagnosis of ischemic and hemorrhagic strokes (5). Afterwards, the point of care test can reduce the time by explaining the on-site diagnosis result to the patient (6).

**Figure 4 ijms-23-08367-f004:**
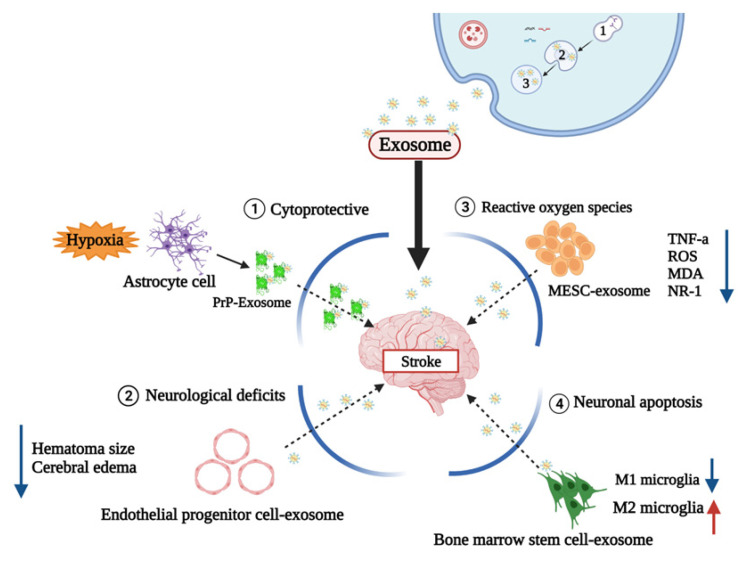
Therapeutic value about stroke mechanism. Mechanism of exosomes as therapeutic agents: (1) Prion proteins are associated with cell defense against hypoxia and damage caused by ischemia (2) Endotheliological progenitor cell derived exosome reduces neurologic definitions (3) Embryonic stem cell exosomes (MESC-exotic) containing curcumin reduce ROS-related genes (4) Exosomes derived from bone marrow stem cells inhibit microglia inflammation and protect microglia from inflammatory neurotoxicity.

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
