# Peer review of "Utility of Exosomes in Ischemic and Hemorrhagic Stroke Diagnosis and Treatment"

_ijms, 2022, doi:10.3390/ijms23158367_

Round 1

Reviewer 1 Report

The article “Utility of exosomes in ischemic and hemorrhagic stroke diagnosis and treatment” by Eun Chae Lee Tae Won Ha, Dong Hun Lee, DongYong Hong, SangWon Park, Ji young Lee, Man Ryul Lee, and JaeSang Oh is attempting to summarize recent findings in the growing field of exosome biology. The theme is very interesting and is of high importance for both fundamental biology and medicine. But this article is weak and needs serious improvement.

Main issue is the article don’t address any specific goal. We see many perspectives for the exosome utilization and don’t see solid facts under these possibilities. Crucial details are missing. First, reference list in the article is incomplete. Second, there is no proofs for the authors conclusions. Third, we need critical presentation of experimental results. The article should be carefully redesigned and rewritten.

Places in the article that require correction are highlighted in yellow.

Line 38. What kind of exosomes demonstrate regenerative potential? Only stem cell exosomes or other type? Exogenous or endogenous? Please, provide references. The article is about regenerative potential of exosomes, so this point should be completed in details.

Lines 38-40. What is the source of nontumorigenic exosomes? Exosomes sometimes promote tumor growth (Hoshino et al., 2015, Nature) and this point should be highlighted in the article.

Line 56. Functional genetic information – this should be illustrated with reference.

Line 61. Apoptotic bodies are of another origin.

Lines 78-81. Please provide reference for experimental results, not review.

Figures in the article are of poor quality. Please provide high resolution figures.

Line 111. Please, provide reference.

Line 112. Neurites is peripheral disease, not central.

Lines 115-120. Epidemiological information is unnecessary for this article.

Line 139 and line 158. Methods section should be completely rewritten. Please, provide comprehensive review of the methods used in exosomes isolation.

Lines 197-199. Please provide reference for experimental results, not review.

Lines 202-203. Reference provide evidence for antiangiogenic properties of exosomes, not proangiogenic.

Line 203. Please, provide reference.

Line 204. Please, correct “intermediate-strength stem cells”.

Line 226. Please, provide references.

Lines 231-233. Relationship between angiogenesis and neurogenesis is non obvious.

Line 236. Please, correct “anatomically activated”.

Lines 238-240. Please, provide reference.

Line 245. Please provide reference for experimental results, not review.

Line 247. Please, correct “epileptic persistence”.

Line 247. Reference 57 provides evidence for exosomes that prevent neurogenesis in the brain. But early in the article we read about pro-neurogenic action of the exosomes. How can we explain it?

Line 249. Please provide reference for experimental results, not review, instead of ref 58.

Line 273. Provided references don’t illustrate author suggestions.

Lines 285-287. Please, provide reference.

Lines 292-294. Please, provide reference.

Lines 307-309. Please, provide reference.

Lines 316-318. Repetition of lines 297-299.

Line 321. Please, provide more appropriate reference.

Lines 323-324. Please, provide reference.

Line 328. Porcine model, not mouse.

Lines 329-331. Please, provide reference.

Lines 332-334. Please, provide reference.

Line 335. Reference is about traumatic brain injury, not stroke.

Line 339. In this context, what is the meaning of “the reticuloendothelial system”?

Line 367. Reference is not about exosomes.

Line 371. Reference is TIA not stroke.

Line 375 and line 378. Reference 85 is not about exosomes.

Line 388. Provided references don’t illustrate author suggestions.

Line 410. Provided references don’t illustrate author suggestions.

Line 430. “of exosomes derived exosomes”.

Line 435. “of higher ducts”.

Lines 445-447. Please, clarify the sentence.

Lines 455-456. Please, clarify “brain reconstruction” and “BBB permeability” in the exosomal context.

Author Response

Dear Managing Editor

Thank you for your attentive comments and we agree with your opinios.

First, we added a more detailed reference list. Second, reference lists are added as footnotes as evidence of the conclusion. Third, it has been modified by adding critical opinions about the experimental results.
These modifications are listed below.

Line 38. What kind of exosomes demonstrate regenerative potential? Only stem cell exosomes or other type? Exogenous or endogenous? Please, provide references. The article is about regenerative potential of exosomes, so this point should be completed in details.

  • According to your opinion, the text has been revised in detail to say that exosomes secreted from various exogenous cells and organs play a potential nerve regeneration role, and references have been added.

We added the results to Line 38-39 on manuscript .

       6. Lapchak, P. A.;  Boitano, P. D.;  de Couto, G.; Marban, E., Intravenous           xenogeneic human cardiosphere-derived cell extracellular vesicles               (exosomes) improves behavioral function in small-clot embolized                 rabbits. Exp Neurol 2018, 307, 109-117.

    1. Hong, S. B.; Yang, H.;  Manaenko, A.;  Lu, J.;  Mei, Q.; Hu, Q., Potential of Exosomes for the Treatment of Stroke. Cell Transplant 2019, 28 (6), 662-670.
    2. Raposo, G.; Stoorvogel, W., Extracellular vesicles: exosomes, microvesicles, and friends. J Cell Biol 2013, 200 (4), 373-83.
    3. Zhang, Z. G.; Buller, B.; Chopp, M., Exosomes - beyond stem cells for restorative therapy in stroke and neurological injury. Nat Rev Neurol 2019, 15 (4), 193-203.

Lines 38-40. What is the source of nontumorigenic exosomes? Exosomes sometimes promote tumor growth (Hoshino et al., 2015, Nature) and this point should be highlighted in the article.

  • It has been demonstrated in many studies that the transfer function of exodots accelerates cancer metastasis. However, a recent study showed that exosomal miRNA-204 decreases the expression of KLF7, thereby inhibiting the Akt/HIF-1α axis, thereby inhibiting EMT, thereby reducing lung cancer migration and invasion. With this possibility, I've written that comment.

We added the results to Line 41 on manuscript .

11. Liu, X. N.; Zhang, C. B.;  Lin, H.;  Tang, X. Y.;  Zhou, R.;  Wen, H. L.; Li, J., microRNA-204 shuttled by mesenchymal stem cell-derived exosomes inhibits the migration and invasion of non-small-cell lung cancer cells via the KLF7/AKT/HIF-1alpha axis. Neoplasma 2021, 68 (4), 719-731.

Line 56. Functional genetic information – this should be illustrated with reference.

  • We identified and added omissions of references to support our opinion. The added references are as follows.

We added the results to Line 58 on manuscript.

  1. S, E. L. A.; Mager, I.;  Breakefield, X. O.; Wood, M. J., Extracellular vesicles: biology and emerging therapeutic opportunities. Nat Rev Drug Discov 2013, 12 (5), 347-57.
  2. Camussi, G.; Deregibus, M. C.;  Bruno, S.;  Cantaluppi, V.; Biancone, L., Exosomes/microvesicles as a mechanism of cell-to-cell communication. Kidney Int 2010, 78 (9), 838-848.
  3. Nazimek, K.; Bryniarski, K.;  Santocki, M.; Ptak, W., Exosomes as mediators of intercellular communication: clinical implications. Pol Arch Med Wewn 2015, 125 (5), 370-380.

Line 61. Apoptotic bodies are of another origin.

  • We agree with your opinion, the mechanism has been further investigated, and the word apoptotic body has been deleted. Figure 1 was also modified.

We added the results to Line 64 on manuscript.

Lines 78-81. Please provide reference for experimental results, not review.

  • We agree with your opinion, and after deleting the review paper, the experimental paper has been modified as a reference.

We added the results to Line 81-83 on manuscript.

  1. Yang, J. L.; Zhang, X. F.;  Chen, X. J.;  Wang, L.; Yang, G. D., Exosome Mediated Delivery of miR-124 Promotes Neurogenesis after Ischemia. Mol Ther-Nucl Acids 2017, 7, 278-287.

Figures in the article are of poor quality. Please provide high resolution figures.

  • We have modified the quality of the aritcle of the figure to high quality according to your comments.

Line 111. Please, provide reference.

  • We identified and added omissions of references to support our opinion. The added references are as follows.

We added the results to Line 124 on manuscript.

  1. D'Anca, M.; Fenoglio, C.;  Serpente, M.;  Arosio, B.;  Cesari, M.;  Scarpini, E. A.; Galimberti, D., Exosome Determinants of Physiological Aging and Age-Related Neurodegenerative Diseases. Front Aging Neurosci 2019, 11, 232.
  2. Du, T.; Yang, C. L.;  Ge, M. R.;  Liu, Y.;  Zhang, P.;  Li, H.;  Li, X. L.;  Li, T.;  Liu, Y. D.;  Dou, Y. C.;  Yang, B.; Duan, R. S., M1 Macrophage Derived Exosomes Aggravate Experimental Autoimmune Neuritis via Modulating Th1 Response. Front Immunol 2020, 11, 1603.

Line 112. Neurites is peripheral disease, not central.

  • We have further confirmed Neurites based on your comments, and have removed them because they do not fit the disease you describe.

We added the results to Line 124 on manuscript.

Lines 115-120. Epidemiological information is unnecessary for this article.

  • We have checked the context of the entire thesis according to your opinion and deleted the epidemiological-related content.

We added the results to Line 124 on manuscript.

Line 139 and line 158. Methods section should be completely rewritten. Please, provide comprehensive review of the methods used in exosomes isolation.

  • We agree with your opinion and have revised the overall context into a comprehensive review format. We wrote objective advantages and disadvantages rather than detailed exosome isolation protocols.

We added the results to Line 150-174 on manuscript.

Lines 197-199. Please provide reference for experimental results, not review.

  • We agree with your comments and have attached references as experimental papers.

We added the results to Line 196 on manuscript.

  1. Li, Y. F.; Ren, L. N.;  Guo, G.;  Cannella, L. A.;  Chernaya, V.;  Samuel, S.;  Liu, S. X.;  Wang, H.; Yang, X. F., Endothelial progenitor cells in ischemic stroke: an exploration from hypothesis to therapy. J Hematol Oncol 2015, 8, 33.
  2. Chen, K.; Li, Y.;  Xu, L. W.;  Qian, Y. G.;  Liu, N.;  Zhou, C. C.;  Liu, J. Y.;  Zhou, L. H.;  Xu, Z.;  Jia, R. P.; Ge, Y. Z., Comprehensive insight into endothelial progenitor cell-derived extracellular vesicles as a promising candidate for disease treatment. Stem Cell Res Ther 2022, 13 (1).

Lines 202-203. Reference provide evidence for antiangiogenic properties of exosomes, not proangiogenic.

  • We agree with your opinion and have revised it to refer to ovide evidence for antiangiogenic properties of exosomes rather than proangiogenic papers.

We added the results to Line 200 on manuscript.

  1. Wang, X.; Huang, W.;  Liu, G.;  Cai, W.;  Millard, R. W.;  Wang, Y.;  Chang, J.;  Peng, T.; Fan, G. C., Cardiomyocytes mediate anti-angiogenesis in type 2 diabetic rats through the exosomal transfer of miR-320 into endothelial cells. J Mol Cell Cardiol 2014, 74, 139-50.

Line 203. Please, provide reference.

  • We agree with your comments and have added references that were not added.

We added the results to Line 202 on manuscript.

  1. Thej, C.; Kishore, R., Unfathomed Nanomessages to the Heart: Translational Implications of Stem Cell-Derived, Progenitor Cell Exosomes in Cardiac Repair and Regeneration. Cells 2021, 10 (7).
  2. Vrijsen, K. R.; Maring, J. A.;  Chamuleau, S. A.;  Verhage, V.;  Mol, E. A.;  Deddens, J. C.;  Metz, C. H.;  Lodder, K.;  van Eeuwijk, E. C.;  van Dommelen, S. M.;  Doevendans, P. A.;  Smits, A. M.;  Goumans, M. J.; Sluijter, J. P., Exosomes from Cardiomyocyte Progenitor Cells and Mesenchymal Stem Cells Stimulate Angiogenesis Via EMMPRIN. Adv Healthc Mater 2016, 5 (19), 2555-2565.

Line 204. Please, correct “intermediate-strength stem cells”.

  • We did further investigations on "intermediate-strength stem cells" based on your comments, but deleted them due to unclear results.

We added the results to Line 201 on manuscript.

Line 226. Please, provide references.

We have confirmed your comments and have further researched and referenced the references.

We added the results to Line 224 on manuscript.

63. Thej, C.; Kishore, R., Unfathomed Nanomessages to the Heart: Translational Implications of Stem Cell-Derived, Progenitor Cell Exosomes in Cardiac Repair and Regeneration. Cells 2021, 10 (7).

72. Chen, S. Y.; Lin, M. C.;  Tsai, J. S.;  He, P. L.;  Luo, W. T.;  Chiu, I. M.;  Herschman, H. R.; Li, H. J., Exosomal 2',3'-CNP from mesenchymal stem cells promotes hippocampus CA1 neurogenesis/neuritogenesis and contributes to rescue of cognition/learning deficiencies of damaged brain. Stem Cells Transl Med 2020, 9 (4), 499-517.

Lines 231-233. Relationship between angiogenesis and neurogenesis is non obvious.

  • We confirm your opinion and confirm that the relationship between angiogensis and neurogenesis is non abvious. Therefore, that paragraph has been deleted.

Line 236. Please, correct “anatomically activated”.

  • We have confirmed your comment and amended it to "activated cerebral endothelial cells". Reference was made to the corresponding additional references.

We added the results to Line 232 on manuscript.

  1. Furutachi, S.; Miya, H.;  Watanabe, T.;  Kawai, H.;  Yamasaki, N.;  Harada, Y.;  Imayoshi, I.;  Nelson, M.;  Nakayama, K. I.;  Hirabayashi, Y.; Gotoh, Y., Slowly dividing neural progenitors are an embryonic origin of adult neural stem cells. Nat Neurosci 2015, 18 (5), 657-65.
  2. Codega, P.; Silva-Vargas, V.;  Paul, A.;  Maldonado-Soto, A. R.;  DeLeo, A. M.;  Pastrana, E.; Doetsch, F., Prospective Identification and Purification of Quiescent Adult Neural Stem Cells from Their In Vivo Niche. Neuron 2014, 82 (3), 545-559.
  3. Wang, X.; Mao, X.;  Xie, L.;  Sun, F.;  Greenberg, D. A.; Jin, K., Conditional depletion of neurogenesis inhibits long-term recovery after experimental stroke in mice. PLoS One 2012, 7 (6), e38932.

Lines 238-240. Please, provide reference.

  • Based on your comments, we have researched and added contextual references.

We added the results to Line 234-238 on manuscript.

  1. Zhang, R. L.; Chopp, M.;  Roberts, C.;  Liu, X.;  Wei, M.;  Nejad-Davarani, S. P.;  Wang, X.; Zhang, Z. G., Stroke increases neural stem cells and angiogenesis in the neurogenic niche of the adult mouse. PLoS One 2014, 9 (12), e113972.
  2. Ohab, J. J.; Fleming, S.;  Blesch, A.; Carmichael, S. T., A neurovascular niche for neurogenesis after stroke. J Neurosci 2006, 26 (50), 13007-16.
  3. Silva-Vargas, V.; Crouch, E. E.; Doetsch, F., Adult neural stem cells and their niche: a dynamic duo during homeostasis, regeneration, and aging. Curr Opin Neurobiol 2013, 23 (6), 935-942.
  4. Robin, A. M.; Zhang, Z. G.;  Wang, L.;  Zhang, R. L.;  Katakowski, M.;  Zhang, L.;  Wang, Y.;  Zhang, C.; Chopp, M., Stromal cell-derived factor 1alpha mediates neural progenitor cell motility after focal cerebral ischemia. J Cereb Blood Flow Metab 2006, 26 (1), 125-34.
  5. Huttner, H. B.; Bergmann, O.;  Salehpour, M.;  Racz, A.;  Tatarishvili, J.;  Lindgren, E.;  Csonka, T.;  Csiba, L.;  Hortobagyi, T.;  Mehes, G.;  Englund, E.;  Solnestam, B. W.;  Zdunek, S.;  Scharenberg, C.;  Strom, L.;  Stahl, P.;  Sigurgeirsson, B.;  Dahl, A.;  Schwab, S.;  Possnert, G.;  Bernard, S.;  Kokaia, Z.;  Lindvall, O.;  Lundeberg, J.; Frisen, J., The age and genomic integrity of neurons after cortical stroke in humans. Nat Neurosci 2014, 17 (6), 801-3.
  6. Long, Q.; Upadhya, D.;  Hattiangady, B.;  Kim, D. K.;  An, S. Y.;  Shuai, B.;  Prockop, D. J.; Shetty, A. K., Intranasal MSC-derived A1-exosomes ease inflammation, and prevent abnormal neurogenesis and memory dysfunction after status epilepticus. P Natl Acad Sci USA 2017, 114 (17), E3536-E3545.

Line 245. Please provide reference for experimental results, not review.

  • In accordance with your opinion, the review paper was deleted and the experimental paper was revised as a reference.

We added the results to Line 242 on manuscript.

  1. Feliciano, D. M.; Zhang, S.;  Nasrallah, C. M.;  Lisgo, S. N.; Bordey, A., Embryonic cerebrospinal fluid nanovesicles carry evolutionarily conserved molecules and promote neural stem cell amplification. PLoS One 2014, 9 (2), e88810.

Line 247. Please, correct “epileptic persistence”.

  • Changed from "epileptic persistence" to "status epilepticus" based on your comments.

We added the results to Line 244 on manuscript.

Line 247. Reference 57 provides evidence for exosomes that prevent neurogenesis in the brain. But early in the article we read about pro-neurogenic action of the exosomes. How can we explain it?

  • Various studies have shown evidence that protecting newborn neurons exerts pro-neurogenic activity. Therefore, it is considered that there is a close relationship between neurogenesis and pro-neurogenic action. Additional relevant references are provided.

We added the results to Line 246 on manuscript.

89. Pieper, A. A.; Xie, S.;  Capota, E.;  Estill, S. J.;  Zhong, J.;  Long, J. M.;  Becker, G. L.;  Huntington, P.;  Goldman, S. E.;  Shen, C. H.;  Capota, M.;  Britt, J. K.;  Kotti, T.;  Ure, K.;  Brat, D. J.;  Williams, N. S.;  MacMillan, K. S.;  Naidoo, J.;  Melito, L.;  Hsieh, J.;  De Brabander, J.;  Ready, J. M.; McKnight, S. L., Discovery of a proneurogenic, neuroprotective chemical. Cell 2010, 142 (1), 39-51.

Line 249. Please provide reference for experimental results, not review, instead of ref 58.

  • Based on your comments, it has been modified as an experimental paper in place of ref 58 .

We added the results to Line 246 on manuscript.

  1. Kim, D. K.; Nishida, H.;  An, S. Y.;  Shetty, A. K.;  Bartosh, T. J.; Prockop, D. J., Chromatographically isolated CD63+CD81+ extracellular vesicles from mesenchymal stromal cells rescue cognitive impairments after TBI. Proc Natl Acad Sci U S A 2016, 113 (1), 170-5.

Line 273. Provided references don’t illustrate author suggestions.

  • We agree with your opinion and have revised it into the same reference as the opinion presented.

We added the results to Line 270 on manuscript.

  1. Gong, Y.; He, Y.;  Gu, Y.;  Keep, R. F.;  Xi, G.; Hua, Y., Effects of aging on autophagy after experimental intracerebral hemorrhage. Acta Neurochir Suppl 2011, 111, 113-7.
  2. He, Y.; Wan, S.;  Hua, Y.;  Keep, R. F.; Xi, G., Autophagy after experimental intracerebral hemorrhage. J Cereb Blood Flow Metab 2008, 28 (5), 897-905.
  3. Pei, X.; Li, Y.;  Zhu, L.; Zhou, Z., Astrocyte-derived exosomes suppress autophagy and ameliorate neuronal damage in experimental ischemic stroke. Exp Cell Res 2019, 382 (2), 111474.

Lines 285-287. Please, provide reference.

  • We agreed to your presentation and checked the sentence, but deleted it after confirming that it was not accurate. However, research results related to Autophagy have been reviewed, and relevant references have been added.

We added the results to Line 282-291 on manuscript.

  1. Zang, J.; Wu, Y.;  Su, X.;  Zhang, T.;  Tang, X.;  Ma, D.;  Li, Y.;  Liu, Y.;  Weng, Z.;  Liu, X.;  Tsang, C. K.;  Xu, A.; Lu, D., Inhibition of PDE1-B by Vinpocetine Regulates Microglial Exosomes and Polarization Through Enhancing Autophagic Flux for Neuroprotection Against Ischemic Stroke. Front Cell Dev Biol 2020, 8, 616590.
  2. Zang, J. K.; Wu, Y. S.;  Su, X. L.;  Zhang, T. Y.;  Tang, X. L.;  Ma, D.;  Li, Y. F.;  Liu, Y. F.;  Weng, Z. A.;  Liu, X. Z.;  Tsang, C. W.;  Xu, A. D.; Lu, D., Inhibition of PDE1-B by Vinpocetine Regulates Microglial Exosomes and Polarization Through Enhancing Autophagic Flux for Neuroprotection Against Ischemic Stroke. Front Cell Dev Biol 2021, 8.
  3. Chen, W.; Wang, H.;  Zhu, Z.;  Feng, J.; Chen, L., Exosome-Shuttled circSHOC2 from IPASs Regulates Neuronal Autophagy and Ameliorates Ischemic Brain Injury via the miR-7670-3p/SIRT1 Axis. Mol Ther Nucleic Acids 2020, 22, 657-672.

Lines 292-294. Please, provide reference.

  • We agree with your comments and have added a reference to that sentence.

We added the results to Line 298 on manuscript.

  1. Zhang, Y.; Khan, S.;  Liu, Y.;  Zhang, R.;  Li, H.;  Wu, G.;  Tang, Z.;  Xue, M.; Yong, V. W., Modes of Brain Cell Death Following Intracerebral Hemorrhage. Front Cell Neurosci 2022, 16, 799753.

Lines 307-309. Please, provide reference.

  • We agree with your comments and have provided additional references to the text.

We added the results to Line 313 on manuscript.

Zhai, K.; Duan, H.;  Wang, W.;  Zhao, S.;  Khan, G. J.;  Wang, M.;  Zhang, Y.;  Thakur, K.;  Fang, X.;  Wu, C.;  Xiao, J.; Wei, Z., Ginsenoside Rg1 ameliorates blood-brain barrier disruption and traumatic brain injury via attenuating macrophages derived exosomes miR-21 release. Acta Pharm Sin B 2021, 11 (11), 3493-3507.

Lines 316-318. Repetition of lines 297-299.

  • We checked the repetition of the sentence you presented, and deleted it as unnecessary repetition.

Line 321. Please, provide more appropriate reference.

  • We agree with your comments and have added appropriate references to those suggested comments.

We added the results to Line 323 on manuscript.

  1. Chen, C. C.; Liu, L.;  Ma, F.;  Wong, C. W.;  Guo, X. E.;  Chacko, J. V.;  Farhoodi, H. P.;  Zhang, S. X.;  Zimak, J.;  Segaliny, A.;  Riazifar, M.;  Pham, V.;  Digman, M. A.;  Pone, E. J.; Zhao, W., Elucidation of Exosome Migration across the Blood-Brain Barrier Model In Vitro. Cell Mol Bioeng 2016, 9 (4), 509-529.

107. Anil, A.; Banerjee, A., Pneumococcal Encounter With the Blood-Brain Barrier Endothelium. Front Cell Infect Microbiol 2020, 10, 590682.

108. Lu, Y.; Chen, L.;  Li, L.; Cao, Y., Exosomes Derived from Brain Metastatic Breast Cancer Cells Destroy the Blood-Brain Barrier by Carrying lncRNA GS1-600G8.5. Biomed Res Int 2020, 2020, 7461727.

Lines 323-324. Please, provide reference.

  • We agree with your comments and have added references to support those comments.

We added the results to Line 326 on manuscript.

  1. Jakubec, M.; Maple-Grodem, J.;  Akbari, S.;  Nesse, S.;  Halskau, O.; Mork-Jansson, A. E., Plasma-derived exosome-like vesicles are enriched in lyso-phospholipids and pass the blood-brain barrier. PLoS One 2020, 15 (9), e0232442.

Line 328. Porcine model, not mouse.

  • We have confirmed your comments. After checking the references for the study, the Porcine model is correct. Therefore, the mouse model was deleted and the "Porcine model" was modified and presented.

We added the results to Line 329 on manuscript.

Lines 329-331. Please, provide reference.

  • We agree with your comments and have added appropriate references.

We added the results to Line 333 on manuscript. 

  1. Pelz, J.; Hartig, W.;  Weise, C.;  Hobohm, C.;  Schneider, D.;  Krueger, M.;  Kacza, J.; Michalski, D., Endothelial barrier antigen-immunoreactivity is conversely associated with blood-brain barrier dysfunction after embolic stroke in rats. Eur J Histochem 2013, 57 (4), 255-261.
  2. Doeppner, T. R.; Herz, J.;  Gorgens, A.;  Schlechter, J.;  Ludwig, A. K.;  Radtke, S.;  de Miroschedji, K.;  Horn, P. A.;  Giebel, B.; Hermann, D. M., Extracellular Vesicles Improve Post-Stroke Neuroregeneration and Prevent Postischemic Immunosuppression. Stem Cells Transl Med 2015, 4 (10), 1131-43.

Lines 332-334. Please, provide reference.

  • We agree with your comments and have added appropriate references to support them.

We added the results to Line 336 on manuscript. 

  1. Wang, L.; Zhang, Z. G.;  Wang, Y.;  Zhang, R. L.; Chopp, M., Treatment of stroke with erythropoietin enhances neurogenesis and angiogenesis and improves neurological function in rats. Stroke 2004, 35 (7), 1732-1737.
  2. Wang, L.; Chopp, M.;  Gregg, S. R.;  Zhang, R. L.;  Teng, H.;  Jiang, A.;  Feng, Y. F.; Zhang, Z. G., Neural progenitor cells treated with EPO induce angiogenesis through the production of VEGF. J Cerebr Blood F Met 2008, 28 (7), 1361-1368.
  3. Zacharek, A.; Chen, J. L.;  Cui, X.;  Li, A.;  Li, Y.;  Roberts, C.;  Feng, Y. F.;  Gao, Q.; Chopp, M., Angiopoietin1/Tie2 and VEGF/Flk1 induced by MSC treatment amplifies angiogenesis and vascular stabilization after stroke. J Cerebr Blood F Met 2007, 27 (10), 1684-1691.

Line 335. Reference is about traumatic brain injury, not stroke.

  • We have confirmed your comments. Since the review paper is about stroke, references on traumatic brain injury have been deleted and appropriate references have been presented again.

We added the results to Line 337 on manuscript. 

  1. Yao, X.; Yang, W. P.;  Ren, Z. D.;  Zhang, H. R.;  Shi, D. F.;  Li, Y. F.;  Yu, Z. Y.;  Guo, Q.;  Yang, G. W.;  Gu, Y. J.;  Zhao, H. R.; Ren, K., Neuroprotective and Angiogenesis Effects of Levetiracetam Following Ischemic Stroke in Rats. Front Pharmacol 2021, 12.

Line 339. In this context, what is the meaning of “the reticuloendothelial system”?

  • We have confirmed your comments. However, the word is also presented in the bibliography. We determined that further clarification was necessary and provided additional references.

We added the results to Line 342 on manuscript. 

Choi, H.; Choi, Y.;  Yim, H. Y.;  Mirzaaghasi, A.;  Yoo, J. K.; Choi, C., Biodistribution of Exosomes and Engineering Strategies for Targeted Delivery of Therapeutic Exosomes. Tissue Eng Regen Med 2021, 18 (4), 499-511.

Smyth, T.; Kullberg, M.;  Malik, N.;  Smith-Jones, P.;  Graner, M. W.; Anchordoquy, T. J., Biodistribution and delivery efficiency of unmodified tumor-derived exosomes. J Control Release 2015, 199, 145-155.

Allen, T. M.; Chonn, A., Large Unilamellar Liposomes with Low Uptake into the Reticuloendothelial System. Febs Lett 1987, 223 (1), 42-46.

Line 367. Reference is not about exosomes.

We have confirmed your comments. Existing references were reconfirmed that they were not related to exosomes and were modified to provide additional references.

We added the results to Line 370 on manuscript. 

    1. Kanninen, K. M.; Bister, N.;  Koistinaho, J.; Malm, T., Exosomes as new diagnostic tools in CNS diseases. Bba-Mol Basis Dis 2016, 1862 (3), 403-410.

Line 371. Reference is TIA not stroke.

  • We agree with your comments. Therefore, references related to TIA were deleted and additional references supporting Stroke were created.

We added the results to Line 375 on manuscript. 

  1. Simak, J.; Gelderman, M. P.;  Yu, H.;  Wright, V.; Baird, A. E., Circulating endothelial microparticles in acute ischemic stroke: a link to severity, lesion volume and outcome. J Thromb Haemost 2006, 4 (6), 1296-302.

Line 375 and line 378. Reference 85 is not about exosomes.

  • We have checked the bibliography of your comments. It was confirmed that it was not a reference for exosome, and additional references on exosome were presented.

We added the results to Line 379 on manuscript. 

127.   Zhou, J.;  Chen, L.;  Chen, B.;  Huang, S.;  Zeng, C.;  Wu, H.;  Chen, C.; Long, F., Increased serum exosomal miR-134 expression in the acute ischemic stroke patients. BMC Neurol 2018, 18 (1), 198.

Line 388. Provided references don’t illustrate author suggestions.

  • We have checked the bibliography of your comments. It has been confirmed that the content is not a reference, and additional appropriate references have been provided.

We added the results to Line 391 on manuscript. 

  1. Chatterjee, V.; Yang, X.;  Ma, Y.;  Wu, M. H.; Yuan, S. Y., Extracellular vesicles: new players in regulating vascular barrier function. Am J Physiol Heart Circ Physiol 2020, 319 (6), H1181-H1196.

Line 410. Provided references don’t illustrate author suggestions.

  • We have checked the bibliography of your comments. It has been confirmed that the content is not a reference, and additional appropriate references have been provided.

We added the results to Line 414 on manuscript. 

  1. Walsh, K. B., Non-invasive sensor technology for prehospital stroke diagnosis: Current status and future directions. Int J Stroke 2019, 14 (6), 592-602.
  2. Harpaz, D.; Eltzov, E.;  Seet, R. C. S.;  Marks, R. S.; Tok, A. I. Y., Point-of-Care-Testing in Acute Stroke Management: An Unmet Need Ripe for Technological Harvest. Biosensors (Basel) 2017, 7 (3).

Line 430. “of exosomes derived exosomes”.

We have confirmed your comments. Instead of the expression "exosome derived exososome", it has been modified as "cells derived exosomes" according to the references.

We added the results to Line 449 on manuscript. 

Line 435. “of higher ducts”.

  • We have confirmed your comments. Instead of the expression "higher ducts", it has been modified to "higher canal" according to the references.

We added the results to Line 449 on manuscript. 

Lines 445-447. Please, clarify the sentence.

  • We agree with your revision comments. So instead of "However, the amount of blood in the blood vessels and blood flow can cause insufficient exosomal concetration in the brian" instead of "However, depending on the blood volume and blood flow in the blood vessels, it can cause a lack of concentration of the administered exosomes." Edited to make it more accurate.

We added the results to Line 464-466 on manuscript. 

Lines 455-456. Please, clarify “brain reconstruction” and “BBB permeability” in the exosomal context.

  • We agree with you. Therefore, the brain reconstruction was modified by changing “brain neuron cell reconstruction” and “BBB permeability” to “target delivery by penetrating the BBB”.

We added the results to Line 475-477 on manuscript. 

Again, thank you for giving us the opportunity to strengthen our manuscript with your valuable comments and queries. We have worked hard to incorporate your feedback and hope that these revisions persuade you to accept our submission.

Sincerely,

Jae Sang Oh

Reviewer 2 Report

This is a very relevant and interesting review article. I have a few suggestions that might increase reader interest and reader understanding of the work.

 The quality of figure 1 needs to be significantly improved

 The molecular mechanisms of cell damage during ischemia should be considered in more detail. Taking into account calcium ions, reactive oxygen species, activation of damaging factors. It is desirable that a figures be prepared in this vein and that it reflects cytoprotective inputs from vesicles. I propose to draw this figure as a final to the entire review.

Limitations on the use of vesicles should be discussed. Association with proapoptotic action.

 The therapeutic window for the use of vesicles must be discussed.

Author Response

Dear Managing Editor

Thank you for your attentive comments and we agree with your opinions.

You have raised an important point. Therefore, we tried to modify the molecular mechanism of cell damage during ischemia in Figure 1. However, it is insufficient to express the various mechanisms, so Figure 4 is additionally presented to show how the mechanisms you suggested work for therapeutic purposes.

Mechanism of exosomes as therapeutic agents. 1. Cytoprotective, 2. Neurological deficits, 3. Reactive oxygen species. 4. Neuronal apoptosis has been summarized, and the contents and references are as follows.

  • We added comments regarding Cytoprotective in lines 437 of the manuscript.

143. Guitart, K.;  Loers, G.;  Buck, F.;  Bork, U.;  Schachner, M.; Kleene, R., Improvement of Neuronal Cell Survival by Astrocyte-derived Exosomes Under Hypoxic and Ischemic Conditions Depends on Prion Protein. Glia 2016, 64 (6), 896-910.

  • We added comments regarding Neurological deficits in lines 438-441 of the manuscript.

144. Kalani, A.;  Chaturvedi, P.;  Kamat, P. K.;  Maldonado, C.;  Bauer, P.;  Joshua, I. G.;  Tyagi, S. C.; Tyagi, N., Curcumin-loaded embryonic stem cell exosomes restored neurovascular unit following ischemia-reperfusion injury. Int J Biochem Cell Biol 2016, 79, 360-369.

  • We added comments regarding Reactive oxygen species deficits in lines 447-453 of the manuscript.

147. Bihl, J. C.; Zhang, C.;  Zhao, Y.;  Xiao, X.;  Ma, X.;  Chen, Y.;  Chen, S.;  Zhao, B.; Chen, Y., Angiotensin-(1-7) counteracts the effects of Ang II on vascular smooth muscle cells, vascular remodeling and hemorrhagic stroke: Role of the NFsmall ka, CyrillicB inflammatory pathway. Vascul Pharmacol 2015, 73, 115-123.

148. Zazulia, A. R.; Diringer, M. N.;  Derdeyn, C. P.; Powers, W. J., Progression of mass effect after intracerebral hemorrhage. Stroke 1999, 30 (6), 1167-73.

  • We added comments regarding Neuronal apoptosis deficits in lines 442-446 of the manuscript.

145. Brifault, C.; Gras, M.;  Liot, D.;  May, V.;  Vaudry, D.; Wurtz, O., Delayed pituitary adenylate cyclase-activating polypeptide delivery after brain stroke improves functional recovery by inducing m2 microglia/macrophage polarization. Stroke 2015, 46 (2), 520-8.

146. Zong, L.; Huang, P.;  Song, Q.; Kang, Y., Bone marrow mesenchymal stem cells-secreted exosomal H19 modulates lipopolysaccharides-stimulated microglial M1/M2 polarization and alleviates inflammation-mediated neurotoxicity. Am J Transl Res 2021, 13 (3), 935-951.

Again, thank you for giving us the opportunity to strengthen our manuscript with your valuable comments and queries. We have worked hard to incorporate your feedback and hope that these revisions persuade you to accept our submission.

Sincerely,

Jae Sang Oh

Round 2

Reviewer 1 Report

Accept in present form

Reviewer 2 Report

Dear authors. Article has been greatly improved. It will be of interest to a wide range of readers. The article is now well perceived on a visual level. I am recommend the article for publication. A small note - I recommend the authors to re-read the article carefully and correct some grammatical errors.